# Nasal Mucosa Exploited by SARS-CoV-2 for Replicating and Shedding during Reinfection

**DOI:** 10.3390/v14081608

**Published:** 2022-07-23

**Authors:** Heng Li, Xin Zhao, Jing Li, Huiwen Zheng, Yurong Zhao, Jinling Yang, Jingxian Zhou, Fengmei Yang, Yanli Chen, Yuanyuan Zuo, Qingrun Lai, Haiting Long, Yanyan Li, Weihua Jin, Haijing Shi, Longding Liu

**Affiliations:** Institute of Medical Biology, Chinese Academy of Medical Sciences and Peking Union Medical College, Kunming 650118, China; liheng@imbcams.com.cn (H.L.); xz479633385@163.com (X.Z.); sola@imbcams.com.cn (J.L.); zhenghuiwen@imbcams.com.cn (H.Z.); yrrogerzhao@gmail.com (Y.Z.); yangjl0117@imbcams.com.cn (J.Y.); zhoujingxian185@163.com (J.Z.); yangfenmei@imbcams.com.cn (F.Y.); chenyanli2019@163.com (Y.C.); zuoyuanyuany@hotmail.com (Y.Z.); lqr19980815@gmail.com (Q.L.); jetlong@imbcams.com.cn (H.L.); lyy110719@imbcams.com.cn (Y.L.); jinweihua@imbcams.com.cn (W.J.); haijingshi@hotmail.com (H.S.)

**Keywords:** SARS-CoV-2, reinfection, Syrian hamsters, nasal mucosa, replicating and shedding

## Abstract

Reinfection risk is a great concern with regard to the COVID-19 pandemic because a large proportion of the population has recovered from an initial infection, and previous reports found that primary exposure to SARS-CoV-2 protects against reinfection in rhesus macaques without viral presence and pathological injury; however, a high possibility for reinfection at the current stage of the pandemic has been proven. We found the reinfection of SARS-CoV-2 in Syrian hamsters with continuous viral shedding in the upper respiratory tracts and few injuries in the lung, and nasal mucosa was exploited by SARS-CoV-2 for replication and shedding during reinfection; meanwhile, no viral replication or enhanced damage was observed in the lower respiratory tracts. Consistent with the mild phenotype in the reinfection, increases in mRNA levels in cytokines and chemokines in the nasal mucosa but only slight increases in the lung were found. Notably, the high levels of neutralizing antibodies in serum could not prevent reinfection in hamsters but may play roles in benefitting the lung recovery and symptom relief of COVID-19. In summary, Syrian hamsters could be reinfected by SARS-CoV-2 with mild symptoms but with obvious viral shedding and replication, and both convalescent and vaccinated patients should be wary of the transmission and reinfection of SARS-CoV-2.

## 1. Introduction

SARS-CoV-2 has been causing the global pandemic of COVID-19 for 2 years since December 2019. Although a variety of vaccines reduce severe cases and death, public health still must face multiple waves of variant virus invasion because the vaccine’s efficacy in controlling infection is limited [1,2,3]. Breakthrough SARS-CoV-2 infection has been noted not only in vaccinated populations [1,2] but also in people who have recovered from COVID-19 [4]. According to the reinfection clinical data, these reinfections can be caused by the original strain or a variant [4,5]. Reports of patients becoming reinfected with COVID-19 because they tested positive for the virus again after discharge have been made. The possibility of reinfection with SARS-CoV-2 is rather high [5], especially reinfection by the Omicron variant [6], and reinfection has been reported in Hong Kong, The Netherlands, Belgium, Ecuador, Israel, Australia, and the United States [7,8,9,10,11]. Thus, the reinfection risk must be monitored with regard to the COVID-19 pandemic because a large proportion of the population has recovered from an initial infection. Importantly, the severity of reinfection episodes is asymptomatic/mild in 75% of patients [12], and reinfection with asymptomatic/mild symptoms threatens public health. In fact, in humans, reinfection with seasonal coronaviruses occurs both naturally and under experimental conditions [13,14]. Of note, animal coronaviruses are also known to cause reinfection, including in hosts with measurable antibodies [15]; moderate/severe initial infections do not necessarily provide enhanced protection against reinfection, although patients with more severe infection have been found to have higher neutralizing antibody titers [12,16]. Understanding the consequences of SARS-CoV-2 reinfection is essential to predicting the course of the COVID-19 pandemic because this knowledge will allow for important insight into the pathophysiology of this new disease and can guide the ongoing vaccine development efforts.

Some previous studies demonstrated that rhesus macaques rechallenged with the identical SARS-CoV-2 strain during the early recovery phase of the initial infection did not show detectable viral loads or histopathological changes in the lungs and extrapulmonary tissues, and viral shedding remained negative during a 2-week intensive detection period for the virus in nasopharyngeal and anal swabs after the rechallenge [17,18]; however, more observations were lacking over a long period. SARS-CoV-2 infection can cause interstitial pneumonia in animals, characterized by hyperemia and edema and the infiltration of monocytes and lymphocytes in alveoli, but the infection status is different in different animals. The Syrian hamster has been demonstrated to be a good model for studying respiratory viruses, including SARS-CoV-2, SARS-CoV, influenza virus, and adenovirus [19,20,21]. Both Syrian hamsters and rhesus monkeys required approximately 2 weeks to transition into the recovery stage [17,18,20,21] and were challenged with the same amount of approximately 10^6^ CCID_50_ SARS-CoV-2 at 1000 g, but the levels of viral shedding and viral loads in various tissues were higher in the infected hamsters. Apparent weight loss occurred in the SARS-CoV-2-infected hamsters, while no obvious weight change was found in the infected rhesus monkeys. With regard to viral shedding, the peak viral load in nasal swabs and pharyngeal swabs was above 10^11^ copies/mL in hamsters, while the peak viral loads were 10^6.5^ copies/mL in rhesus monkeys. The lungs showed a viral load of 10^10^ copies/g and a high titer of 10^5^–10^7^ CCID_50_/g in hamsters, while the peak viral load was just 2 × 10^7^ copies/g in rhesus monkeys. In addition, the mean serum neutralizing antibody titer was approximately 1:427 at 14 dpi in infected animals, while the titer was just 1:16–1:32 in infected rhesus monkeys. Interestingly, Syrian hamsters are susceptible to reinfection, and the known data mainly demonstrated shedding in the upper respiratory tracts [22,23]. Here, we used Syrian hamsters to investigate the clinical and infectious features in the tissues in reinfection.

Most of the infected Syrian hamsters underwent the early (2 dpi), severe (4 dpi), and start of recovery (7 dpi) phases, and no infectious virus was detected after 7 dpi, although high copies of viral RNA were continuously detected in the nasal washes for 14 days. All hamsters returned to their original weight within 14 days [20,21]. In this study, three groups of Syrian hamsters were infected with SARS-CoV-2, including those with a primary infection, those rechallenged at 14 dpi after the primary infection, and those rechallenged at 21 dpi after the primary infection. We found that Syrian hamsters were reinfected by SARS-CoV-2 during the initial stage of recovery from the primary infection. Serious pathological damage to the lung appeared during the primary infection, but no enhanced lung damage appeared during reinfection after recovery. Importantly, obvious and continuous viral shedding was found in the upper respiratory tracts, and SARS-CoV-2 replicated in the upper respiratory tracts but did not proliferate in the lung during reinfection. The neutralization abilities of the antibodies in the serum were high and increased quickly during reinfection, and neutralized antibodies may play important roles in the resistance to reinfection; however, they could not prevent reinfection.

## 2. Materials and Methods

### 2.1. Animals and Biosafety

This project included three groups: a primary infection group, a secondary infection group at 14 dpi after the primary infection, and a secondary infection group at 21 dpi after the primary infection, and 55 Syrian hamsters were sacrificed (Figure 1A). The data from 0 dpi to 10 dpi in these three groups were gathered and analyzed. The weight changes, viral loads, viral replication, and viral distribution in various tissues were detected by real-time PCR, IHC, RNAscope-IF codetection, and ultrathin-section-transmission electron microscopy (TEM); pathological damage was detected by hematoxylin and eosin (HE) staining, neutralizing antibody titers in the serum, and the mRNA levels of cytokines and chemokines in nasal mucosa and lungs were detected and analyzed. All work with infectious SARS-CoV-2 was performed with approval under Biosafety Level 3 (BSL3) and Animal Biosafety Level 3 (ABSL3) conditions by the Institutional Biosafety Committee of the Institute of Medical Biology (IMB) in Kunming National High-level Biosafety Primate Research Center.

### 2.2. Viruses

The native viral strain SARS-CoV-2-KMS1/2020 (GenBank accession number: MT226610.1) and a variant of SARS-CoV-2 with D614G (SARS-CoV-2/human/USA/Kunming_kms-6/2020, MW264424.1) were isolated from sputum collected from a COVID-19 patient by the Chinese Academy of Medical Sciences (IMBCAMS) and propagated and tittered on Vero cells in DMEM (Sigma, St. Louis, MO, USA). The stock viruses were frozen at −80 °C and prepared for the following experiments.

### 2.3. Ultrathin Section-Transmission Electron Microscopy (TEM) Detection of SARS-CoV-2 Virions

The fresh tissues were fixed in 2.5% electron microscopy-grade glutaraldehyde, rinsed with 0.2 M sodium cacodylate buffer for one week, and then fixed in 1% osmium tetroxide (Nanjing Zhongjingkeyi Technology, Co., Ltd., Nanjing, China) for 2 h. The tissues were rinsed with 0.2 M sodium cacodylate buffer three times and overnight. Then, the tissues were dehydrated through a graded series (30%, 50%, 70%, and 90%) of ethanol dilutions, 90% acetone, and pure acetone and processed for Epon embedding (Sigma-Aldrich, St. Louis, MO, USA; Merck Millipore, Burlington, MA, USA). Ultrathin sections (60 nm) were stained with uranyl acetate and lead citrate (Nanjing Zhongjingkeyi Technology, Co., Ltd.) and were then observed using an H-7650 electron microscope (HITACHI, Ltd., Tokyo, Japan).

### 2.4. Viral Challenge

Fifty-two Syrian hamsters were challenged with 1 × 10^5^ CCID_50_ SARS-CoV-2 in the primary infection, and 18 Syrian hamsters at 14 dpi of primary infection were rechallenged with the same isolate of SARS-CoV-2 in the same amount; the other 18 Syrian hamsters at 21 dpi of primary infection were rechallenged with the same isolate of SARS-CoV-2 in the same amount; the remaining 4 Syrian hamsters were not rechallenged as the negative control of the rechallenge group and were sacrificed at 31 dpi after primary infection.

### 2.5. Viral Load Detection

RNA from oropharyngeal swabs, 100 µL nasal washes, and 100 mg homogenized tissue was extracted using TRIzol reagent (Tiangen, Beijing, China) in 20 µL RNA-free water and 2 µL total RNA was detected by RT-real-time PCR. The primers and probe used were E_Sarbeco_F: 5′-ACAGGTACGTTAATAGTTAATAGCGT-3′; E_Sarbeco_R: 5′-ATATTGCAGCAGTACGCACACA-3′; and E_Sarbeco-P: 5′-ACACTAGCCATCCTTACTGCGCTTCG-3′. For the quantification of viral RNA, a standard curve was generated using 10-fold dilutions of RNA standard, and the standard curve was y = −0.2795x + 10.882. In the detection, sgRNA (subgenomic mRNA) and the corresponding gRNA (genomic mRNA) region share the same probe and reverse primer but differ in the forward primers. The sgRNA-targeting forward primer is located in the 5′ UTR (upstream of the leader-body TRS junction), whereas the cognate gRNA-targeting forward primer is located upstream of the body TRS and coding region [24]. The primers and probe used for sgRNA-E detection were sgLead-CoV-2-F: 5′-CGATCTCTTGTAGATCTGTTCTC-3′, sgRNA-E-R: 5′-ATATTGCAGCAGTACGCACACA-3′, sgRNA-E-probe: 5′-ACACTAGCCATCCTTACTGCGCTTCG-3′. For quantification of the sgRNA-E gene, a standard curve was generated using 10-fold dilutions of RNA standard, and the standard curve was y = −0.2544x + 9.9106.

### 2.6. Infectivity Tests of the Virus Shedding from the Upper Respiratory Tract

#### 2.6.1. Virus Titer Detection in Vero Cells

The virus titers were determined by a microdose cytopathogenic efficiency (CPE) assay. Mixtures of tenfold serially diluted tissue samples (from 10^0^ to 10^−7^) were incubated with 10^4^ Vero cells in 96-well culture plates. After 5 days of culture in a 5% CO_2_ incubator at 37 °C, the cells were checked for the presence of a CPE under a microscope. The virus titers were calculated by Karber’s methods. The equation was log_10_ of CCID_50_/0.1 mL = L + d(S − 0.5), L was the logarithm of the lowest dilution multiple, d was the coefficient of dilution (class interval), and S was the sum of the cytopathic ratios.

#### 2.6.2. The Infectivity Test in Healthy Syrian Hamsters

The nasal washes at 3 dpi of reinfection were captured in 100 µL sterile water, and healthy Syrian hamsters were infected by all of the nasal washes through a nasal drip. Weight loss and viral loads were observed.

### 2.7. Neutralization Assay

Neutralization was measured in a formally validated assay that utilized two isolates of SARS-CoV-2, SARS-CoV-2-KMS1/2020 and SARS-CoV-2/human/USA/Kunming_kms-6/2020. First, 100 CCID_50_ of the virus was incubated with 50 μL per well of 12 twofold serial dilutions of serum samples in duplicate for 1 h at 37 °C in 96-well plates, and four parallel wells were conducted. Cells were added to 10^4^ cells in 100 μL of growth medium per well. After 5–7 days of incubation, the neutralization assay could be judged according to CPE. Serum samples were heat-inactivated for 30 min at 56 °C prior to the assay. The neutralization assays were calculated according to the Reed–Muench method.

### 2.8. Histopathology

The specimens were fixed in 10% formalin for more than one week, and then the samples were fixed in 10% formalin for 2 h, 1 h in 70% ethanol, 1 h in 80% ethanol, 1 h in 90% ethanol, 1 h in 95% ethanol 3 times, 1 h in xylene, 30 min in xylene, 30 min in paraffin, and 1 h in paraffin twice. After slicing, the sections of paraffin-embedded tissue were deparaffinized in xylene, rehydrated in a graded series of ethanol, and rinsed with double-distilled water; then, hematoxylin was added for 15 min, water was added for 1 min, 1% HCl in ethanol was added for 5 s, water was added for 1 min, ammonium hydroxide was added for 10 s, water was added for 1 min, 0.5% eosin was added for 30 s, 75% ethanol was added for 10 s, 95% ethanol was added for 10 s twice, ethanol was added for 10 s twice, and xylene was added for 10 s twice.

### 2.9. Immunohistochemistry (IHC)

The sections of paraffin-embedded tissue were deparaffinized in xylene, rehydrated in a graded series of ethanol, and rinsed with double-distilled water. The sections were incubated with rabbit anti-N antigen of SARS-CoV-2 (Sino Biological, Beijing, China) for 1 h after heat-induced epitope retrieval. Antibody labeling was visualized by the development of DAB. Digital images were captured and evaluated by a histological section scanner (Pannoramic MIDI, 3D HISTECH, Budapest, Hungary).

### 2.10. RNAscope-Immunofluorescence (IF) Codetection

The RNAscope-Immunofluorescence (IF) codetection was detected using the RNAscope^®^ Multiplex Fluorescent v2 Assay combined with immunofluorescence-integrated co-detection (ACD). Paraffin-embedded tissue sections were labeled with an anti-SARS-CoV-2 N protein antibody (Sino Biological) at a 1:500 dilution overnight at 4 °C. Then, ISH probes, including V-nCoV2019-S (ACD) and V-nCoV2019-S-sense (ACD), were hybridized to RNA, followed by the amplification of the signal operation, and the RNAscope^®^ Multiplex Fluorescent v2 Assay was run to visualize SARS-CoV-2 N protein antigens by FITC-conjugated goat anti-rabbit IgG (Abcam, ab6717, Cambridge, UK) at a 1:500 dilution. The images were captured by a Leica TCS SP8 laser confocal microscope.

### 2.11. Inflammatory Cytokine and Chemokine Quantifications by RT-Real-Time PCR

The mRNA abundance of inflammatory cytokines and chemokines in the nasal mucosa and lungs was detected by RT-real-time PCR and normalized to the β-actin housekeeping gene. Eight types of inflammatory cytokines and chemokines were measured: RANTES, IFN-α, MIP-1-α, IFN-γ, IL-6, IP-10, TNF-α, and IL-1-β. The primer sequences are presented in Appendix A.

### 2.12. ELISA

The levels of IgA and IgG in nasal mucosa, lung, and serum were detected by a Syrian hamster IgA or IgG ELISA detection kit (MEIMIAN, Yancheng, China). One hundred milligrams of nasal mucosa and lung tissue were homogenized in 500 µL of PBS and diluted twentyfold, and the sera were diluted one-thousandfold. The diluted samples were added to the precoated antibody plate for half an hour at 37 °C, and in the end, the data were detected at OD450.

### 2.13. Statistical Analysis

The data were analyzed by one-way ANOVA using SPSS PASW statistical software v.18.0. * 0.01 < *p* ≤ 0.05, ** 0.001 < *p* ≤ 0.01, and *** *p* ≤ 0.001.

## 3. Results

### 3.1. Specific Neutralizing Antibodies Were Promoted during Reinfection but Could Not Control SARS-CoV-2 Shedding from the Upper Respiratory Tract

During the primary infection, the hamster weights decreased from 4 dpi to 7 dpi and were lowest at 5 dpi, but during reinfection, their weights continued to increase after rechallenging with SARS-CoV-2 (Figure 1B). Obvious viral shedding was found in the nasal washes and oropharyngeal swabs, and shedding lasted continuously for 10 dpi during reinfection (Figure 1C and Appendix A). We further found the infectivity of the virus from upper respiratory tract shedding in cell culture (Figure 1D) and in hamster infection tests from the nasal washes (Appendix A). The virus titers on the nasal shedding were obvious at 2 dpi of reinfection (Figure 1D). Meanwhile, we did not observe obvious morphological changes in viral particles by TEM in the shedding of nasal washes between the primary infection and reinfection (Figure 1E). We sequenced the SARS-CoV-2 S1 gene in the shedding from the upper respiratory tract, and we found no difference in the S1 gene sequence of SARS-CoV-2 from the nasal washes between the primary infection and reinfection (Appendix A). In addition, the shedding levels between the two reinfection groups were similar and lower than those in the primary infection group (Figure 1C,D and Appendix A). Neutralization antibodies were detected from 5 dpi and reached 1:100 to 1:251 on 14 or 21 dpi against different infectious isolates, KMS1/2020 (native isolates) (Figure 1E) and USA/Kunming_kms-6/2020 (isolate with D614G) (Figure 1F). During secondary infection, the levels of the neutralizing antibody response increased quickly, peaking at approximately 1:608 at 10 dpi. According to the viral shedding, we speculated that the high level of neutralizing antibodies in serum could not prevent viral infection and proliferation in the nasal mucosa.

### 3.2. Replication of SARS-CoV-2 Was Observed in the Upper Respiratory Tracts during Reinfection with SARS-CoV-2 but Not Obviously in the Lungs

The sgRNA-E gene and gRNA-E gene were detected in various tissues, including the upper respiratory tract (nose: the tissues with nares and no cartilage, and nasal mucosa: the mucous tissue attached to cartilage), lower respiratory tract (trachea and lungs), bulbus olfactorius, cerebrum, and jaw (Figure 2), indicating SARS-CoV-2 replication. The replication of SARS-CoV-2 depends on the transcription of negative-sense RNA intermediates that serve as templates for the synthesis of gRNA and multiple different sgRNAs [24], and the sgRNA provides evidence of replicative intermediates of the virus, rather than residual viral RNA [25]. Importantly, during reinfection, sgRNA could only be detected in the upper respiratory tract, while no sgRNA was detected in the lower respiratory tracts, and many sgRNAs at 3 dpi were detected in the two reinfection groups (Figure 2A, Appendix A). We found that increasing viral loads were only present in the upper respiratory tracts, while the viral loads decreased slowly in the lower respiratory tracts from 0 dpi to 10 dpi (Figure 2B, Appendix A). During primary infection, the dynamics of viral loads and viral replication were basically the same, and the levels of gRNA and sgRNA in each tissue were highest at 3 dpi (Figure 2). At the same time, viral loads were detected at 14 and 21 dpi in various tissues; however, few or no sgRNAs could be detected in the upper respiratory tracts, and no sgRNA could be detected in the lower respiratory tract and other tissues (Figure 2). Syrian hamsters could be reinfected by SARS-CoV-2 with mild symptoms or asymptomatic in the stage of recovery from primary infection; however, obvious viral shedding and replication were observed in the upper respiratory tract, and no viral replication was observed in the lower respiratory tract (Figure 1 and Figure 2, Appendix A). We speculated that viral replication in the upper respiratory tracts would be the source of viral shedding during reinfection.

### 3.3. SARS-CoV-2 Replication and Virion Particles Were Mainly Present in the Upper Respiratory Tracts during Reinfection

The distribution of viral particles in the nasal mucosa of hamsters demonstrated SARS-CoV-2 replication during reinfection. SARS-CoV-2 N-proteins were detected by IHC (Figure 3A) and IF (Figure 4), we revealed S-mRNA-positive strands and S-mRNA-negative strands by RNAscope (Figure 4), and we also observed virions by TEM (Figure 3B). Viral replication was confirmed by RNAscope and viral particle assembling observation besides sgRNA detection. At 3 dpi, N-protein, SARS-CoV-2 S-mRNA-positive strands, S-mRNA-negative strands, and virions were present in the local nasal mucosa during reinfection. Comparably, we found that large amounts of the N-proteins, SARS-CoV-2 S-mRNA-positive strands, S-mRNA-negative strands, and virions were distributed in most areas of the nasal mucosa during primary infection at 3 dpi. While SARS-CoV-2 viruses were still detected in the necrotic area at 7 dpi of the primary infection, replication was not obvious during reinfection (Figure 3 and Figure 4). In the lower respiratory tract (Figure 5 and Figure 6), except for a few N-proteins and SARS-CoV-2 S-mRNA-positive strands being present in the lung, no S-mRNA-negative strand was detected at 3 and 7 dpi of reinfection. However, we observed large amounts of the N-proteins, SARS-CoV-2 S-mRNA-positive strands, and negative strands distributed in most alveolar cells during primary infection. In addition, no obvious SARS-CoV-2 virion was observed in the lung in any group.

### 3.4. No Enhanced Pathological Changes in the Lung during the Reinfection of Hamsters Recovering from Severe Lung Damage from the Primary Infection

Severe damage was observed from 3 dpi to 7 dpi during primary infection, and the pathological damage to the lungs began to improve at 7 dpi; however, some damage persisted until 10 dpi. At 5 dpi, large areas of alveolar septal thickening with some substantial lesions as well as large areas of alveolar structure disappearance were found. Along with alveolar cell degeneration necrosis, numerous inflammatory cell infiltrations were found. Similar to the bronchus and bronchioles, numerous mucosal cells were detached, and part of the lumen was blocked. No severe damage to the pulmonary vasculature occurred during SARS-CoV-2 infection, and edema developed locally with a small amount of inflammatory cell infiltration. In the two reinfection groups, slight pathological damage was always present from 0 dpi to 10 dpi, but no enhanced damage appeared after recovery from the primary infection, and the damage slowly improved during reinfection (Figure 7). Together with the results of the neutralizing antibody response, we speculated that sera protection was the main benefit for lung recovery and resulted in symptom relief from COVID-19.

### 3.5. Cytokine/Chemokine Responses in the Nasal Mucosa during Reinfection Were Similar to Those during Primary Infection

To better understand the immunopathogenesis of SARS-CoV-2 infection and reinfection, we investigated the mRNA expression levels of cytokines and chemokines in the nasal mucosa and lung during SARS-CoV-2 infection (Figure 8). The mRNA levels of infection-related cytokines and chemokines were increased in the nasal mucosa during the primary infection and reinfection, but they responded in the lung only during primary infection. In the primary infection, the mRNA levels of cytokines and chemokines including RANTES, IFN-α, MIP-1-α, IFN-γ, IL-6, IP-10, TNF-α, and IL-1-β were increased in the nasal mucosa (Figure 8A) and lung (Figure 8B), except IL-1-β, which was not increased in the lung (Figure 8B). The cytokine and chemokine mRNA levels were highest at 3 dpi or 5 dpi, except for the abundance of TNF-α in the nasal mucosa, which was highest at 7 dpi. These results suggested a host response to SARS-CoV-2 infection, indicating viral invasion and replication. The expression of most cytokines and chemokines recovered to normal, except for the mRNA abundance of TNF-α and IL-1-β in the nasal mucosa, which was still high at 10 dpi. Notably, nearly the same pattern of cytokine/chemokine responses was found in the nasal mucosa during reinfection. The RANTES, IFN-α, MIP-1-α, and IFN-γ mRNA levels during reinfection were similar to the levels in the primary infection group; the levels of IL-6, IP-10, TNF-α, and IL-1-β were increased during reinfection but were lower than those during primary infection. However, few responses were detected in the lung during the second infection, except the levels of TNF-α and MIP-1-α, which increased after reinfection. In addition, no obvious change in the level of IgA in the nasal mucosa, lung, and serum was found by ELISA during the whole infection or reinfection (Figure 8C). These findings suggest the important roles played by cytokine/chemokine responses in inflammatory responses and tissue injuries but also imply mild symptoms in the lung with stealth viral replication in the upper respiratory tract during reinfection.

## 4. Discussion

Current data from published cases raise the possibility of SARS-CoV-2 reinfection [12], which may be more transmissible and usually displays stealth in patients. Reinfection with SARS-CoV-2 displayed the characteristics of asymptomatic/mild COVID-19, making notification more important than ever. Meanwhile, the pathophysiology and immunopathogenesis of SARS-CoV-2 reinfection are also essential to predicting the course of the COVID-19 pandemic. Nonhuman primates [18,26], ferrets [27], Ace2-transgenic mice [28], and Syrian hamsters [21] have been reported as animal models for SARS-CoV-2 study, but their reinfection statuses are different. Our results indicated that the Syrian hamster is a suitable animal model for investigating reinfection by SARS-CoV-2, with few injuries in the lung but viral shedding in the nasal mucosa.

Consistent with previous data [20,21], during the primary infection, the infected Syrian hamsters started to recover after 7 dpi. At 14 days post-primary infection, no infectious virus was detected. Therefore, we set 14 dpi and 21 dpi after the primary infection as the two initial time points for evaluating reinfection progress in the recovery stages. In these two reinfection groups, viral replication was confirmed by sgRNA and RNAscope in the nasal mucosa, and viral particle assembling was observed under TEM. Importantly, in our manuscript, the sgRNA levels detected by real-time PCR were consistent with the SARS-CoV-2 S-negative levels detected by RNAScope in the nasal mucosa and lung during primary infection and reinfection, and both the sgRNA and S-negative strands represent SARS-CoV-2 replication (Figure 2, Figure 4 and Figure 6). Although fewer virions were observed in the nasal mucosa than in primary infection, this result suggested that viral replication in the upper respiratory tract would be the source of viral shedding during reinfection.

Importantly, we found that there was infectivity of the virus shedding from the upper respiratory tract, and Vero cells were cytopathic with high titer levels (Figure 1D). Healthy Syrian hamsters were infected by nasal washes with increasing viral loads in the nasal mucosa and lung and weight loss (Appendix A), and the pathological damage in the lung was similar to that seen during primary infection (not shown). In our previous experiments, Syrian hamsters were challenged with 10^4^ or 10^5^ CCID_50_ SARS-CoV-2 per 100 g, and we found that the total levels of viral load, viral distribution, injuries in various tissues, and weight loss were similar between the 10^4^ and 10^5^ CCID_50_-challenged groups; thus, we speculated that the 10^5^ CCID_50_-challenged dose was high, and a higher dose would not cause much damage, especially in the reinfection groups. Mild infection in the upper respiratory tract was dominant, with a high level of neutralizing antibody in the serum. In addition, we found that the SARS-CoV-2 shedding in the nasal washes during reinfection was not an escape mutant, and there were no differences in the S1 gene sequence of SARS-CoV-2 from the nasal washes between the primary infection and reinfection (Appendix A). We did not observe morphological changes in viral particles by TEM in the nasal mucosa (Figure 3B) and nasal washes (Figure 1E).

Previous reports have shown that primary SARS-CoV-2 exposure protected against subsequent reinfection in rhesus macaques by enhanced neutralizing antibody and immune responses [17], and the presence of neutralizing antibodies could prevent infection and disease (such as lung pathology) and attenuate viral replication in the airway epithelia [29,30]. In our results, no obvious pathological damage appeared in the lungs of the reinfected hamsters, indicating that neutralizing antibodies function to prevent lung damage. However, these neutralizing antibodies in serum were not sufficient to prevent viral infection in the nasal mucosa, in which viral replication and shedding were displayed continuously with lower copies. Notably, the levels of IgA in the nasal mucosa, lung, and serum did not respond during infection and reinfection, and we speculated that the mucosal immune response may not play a role against SARS-COV-2 infection. Serum protection was the main benefit for lung recovery and resulted in symptom relief from COVID-19.

In patients infected with SARS-CoV-2 [31,32], higher levels of the proinflammatory cytokines IL-1β, IL-6, IL-8, IL-13, IL-17, RANTES, and IFN-γ were found, and these increased cytokines and chemokines mediated the infection immunopathogenesis and played important roles in the progression of COVID-19. In our results, cytokines and chemokines also responded in the nasal mucosa and lung during SARS-CoV-2 infection, and their mRNA levels obviously increased in the nasal mucosa but few in the lung during reinfection, which was consistent with the mild phenotype. In the second round of SARS-CoV-2 infection in hamsters, we found the same pattern of cytokine/chemokine responses, including elevated RANTES, IFN-α, MIP-1-α, and IFN-γ mRNA levels, in the nasal mucosa as in primary infection. Meanwhile, the mRNA levels of IL-6, IP-10, TNF-α, and IL-1-β were increased during reinfection but were lower than those during primary infection. These cytokines/chemokines represent inflammatory cell migration and exhibit a variety of proinflammatory activities during infections [32,33,34,35,36]. We found that, in line with the viral replication kinetics, the chemokine/cytokine mRNA profile in the lungs of the SARS-CoV-2-infected animals also exhibited a time-dependent trend of gene expression peaking at 3 dpi or 5 dpi and resolving at 7 dpi or 10 dpi. During the second infection, the levels of viral loads were obviously lower than in the primary infection, but the shedding period lasted for more than 10 days, suggesting that cytokine/chemokine responses played important roles in the inflammatory responses and tissue injuries, but also implied mild symptoms in the lung with stealth viral replication in the upper respiratory tract during reinfection.

In our research, we chose the SARS-CoV-2-KMS1/2020 strain for reinfection research. First, SARS-CoV-2-KMS1/2020 was the native strain of SARS-COV-2 and the most common strain in the early stage of COVID-19. Importantly, SARS-CoV-2-KMS1/2020 had a high fatality rate, and reinfection by SARS-CoV-2-KMS1/2020 is worthy of attention. Second, most COVID-19 vaccines that are currently in circulation were prepared from the native strain, therefore, research on SARS-CoV-2-KMS1/2020 reinfection would be useful in the immunological and protective evaluation of vaccines. In addition, reinfections were validated in homologous and heterologous VOC variants, including B.1.1.7 (Alpha), B.1.351 (Beta), and Delta variants [22,23]. Importantly, we demonstrated that nasal mucosa is exploited by SARS-CoV-2 for replication and shedding during reinfection.

In summary, Syrian hamsters could be reinfected by SARS-CoV-2 with no severe symptoms. However, obvious viral shedding was found in nasal washes, and the nasal mucosa was exploited by SARS-CoV-2 for replication and shedding during reinfection. This type of reinfection may be more transmissible and usually displays stealth in people who have recovered from COVID-19 or the vaccinated population. In the current pandemic, with the continuing increase in COVID-19 infection cases [5], both convalescent and vaccinated patients should be wary of the transmission and infection of SARS-CoV-2.

## Figures and Tables

**Figure 1 viruses-14-01608-f001:**
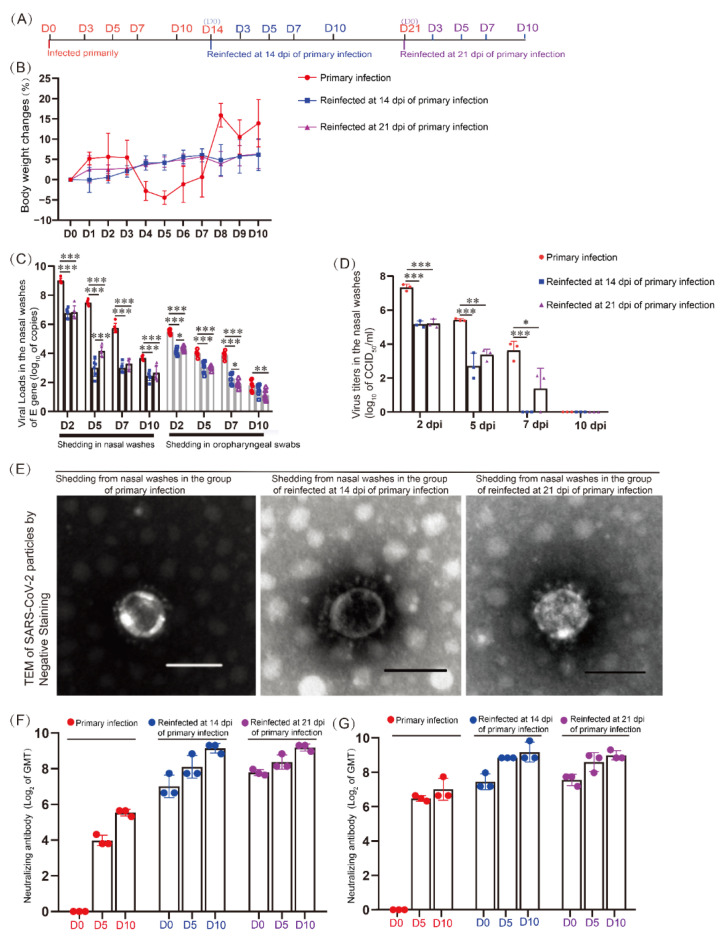
Specific neutralizing antibodies were promoted during reinfection but could not control SARS-CoV-2 shedding from the upper respiratory tract. (**A**) Experimental progress chart. This project included three groups that were infected primarily and infected secondarily at 14 dpi and 21 dpi after primary infection, and the data from 0 dpi to 10 dpi in these three groups were gathered and analyzed. (**B**) Weight changes of the golden hamsters after SARS-CoV-2 infection and reinfection. *n* = 6. (**C**) Viral shedding in 1 mL nasal washes, and viral shedding in oropharyngeal swabs. *n* = 6. (**D**) The SARS-CoV-2 virus titers in nasal washes were detected during primary infection and reinfection. The data are shown as log_10_ of CCID_50_ per 1 mL nasal washes. (**E**) Transmission electron microscopy detection of SARS-CoV-2 virions in the shedding from nasal washes by Negative Staining. The scale was 100 nm. (**F**) The levels of neutralizing antibodies to SARS-CoV-2/human/CHN/KMS1/2020 (native isolate) in the serum of golden hamsters with primary infection and reinfection. (**G**) The levels of neutralizing antibodies to SARS-CoV-2/human/USA/Kunming_kms-6/2020 (D614G variant) in the serum of golden hamsters with primary infection and reinfection. *n* = 3 in every group. The GMT on the *y*-axis is the geometric mean with geometric SD, and the data are shown as log_2_ values. * 0.01 < *p* ≤ 0.05, ** 0.001 < *p* ≤ 0.01, and *** *p* ≤ 0.001.

**Figure 2 viruses-14-01608-f002:**
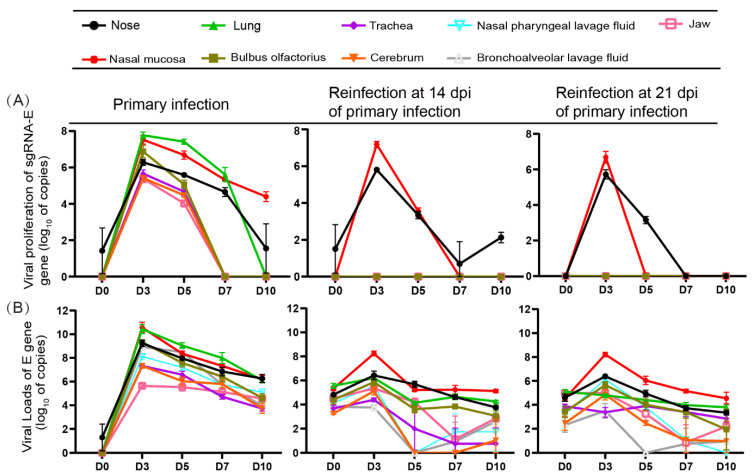
Obvious viral replication was observed in the upper respiratory tract during reinfection, while no viral replication was observed in the lower respiratory tract. (**A**) The levels of the sgRNA-E gene were detected in 1 g of the nose, nasal mucosa, lung, trachea, bulbus olfactorius, cerebrum, and jaw, which represent the areas of viral replication. *n* = 3. (**B**) The levels of the virus load-E gene were detected in 1 g of the nose, nasal mucosa, lung and trachea, bulbus olfactorius, cerebrum, jaw, nasal pharyngeal lavage fluid, and bronchoalveolar lavage fluid. *n* = 3. The levels are shown as log_10_ of copies.

**Figure 3 viruses-14-01608-f003:**
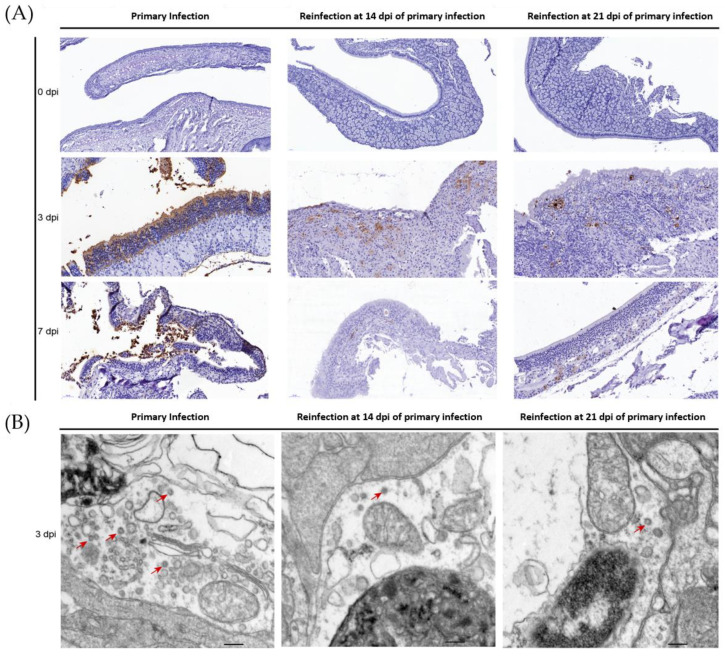
Viral distribution and virions were present in the nasal mucosa during reinfection, while higher levels of viral distribution and virions were observed during primary infection. (**A**) The distribution and abundance of SARS-CoV-2 in the nasal mucosa were detected by SARS-CoV-2-N protein-IHC, and the nasal mucosa was detected at 0 dpi, 3 dpi, and 7 dpi in the primary and secondary infection groups. (**B**) SARS-CoV-2 virions in nasal mucosa were detected by ultrathin section TEM. The scale is 200 nm.

**Figure 4 viruses-14-01608-f004:**
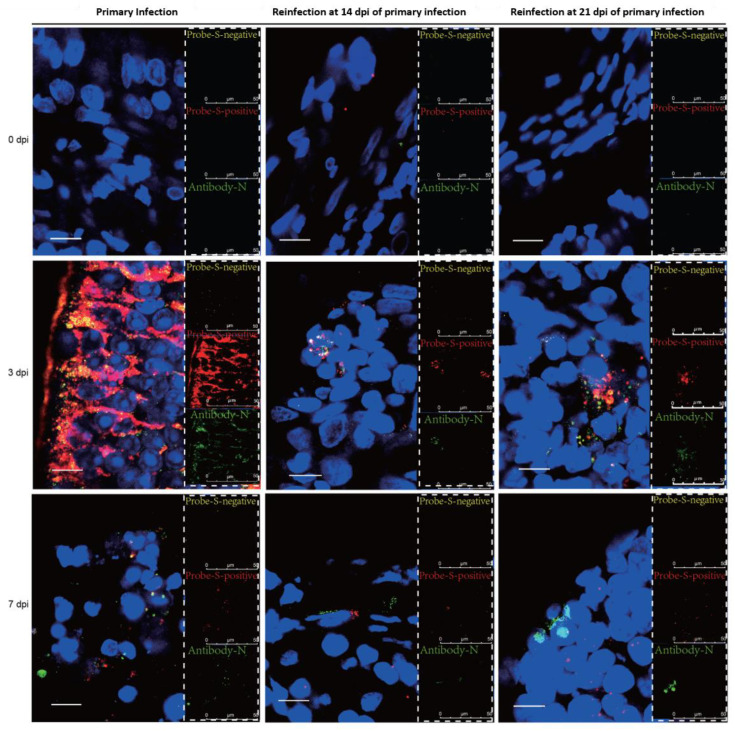
Viral distribution and replication were present in the nasal mucosa in the reinfection, while higher levels of viral replication were observed during primary infection. The distribution and replication of SARS-CoV-2 S-mRNA and N-protein in the nasal mucosa were detected by RNAscope-IF codetection and were detected at 0 dpi, 3 dpi, and 7 dpi in the primary and secondary infection groups. Yellow represents the S-mRNA-negative strand, and the signals represent the level of SARS-CoV-2 replication; red represents the S-mRNA-positive strand, and the signals represent the level of SARS-CoV-2 abundance; green represents the N-protein, and the signals represent the level of SARS-CoV-2 abundance. The scale is 10 µm.

**Figure 5 viruses-14-01608-f005:**
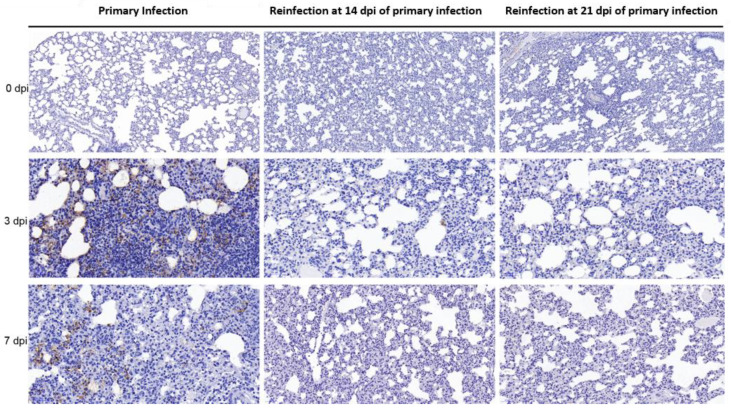
Little viral load was present in the lung in the reinfection, while high levels of SARS-CoV-2 were observed in the primary infection. The distribution and viral load of SARS-CoV-2 in the lung were detected by SARS-CoV-2-N protein-IHC, and the lungs were detected at 0 dpi, 3 dpi, and 7 dpi in the primary and secondary infection groups.

**Figure 6 viruses-14-01608-f006:**
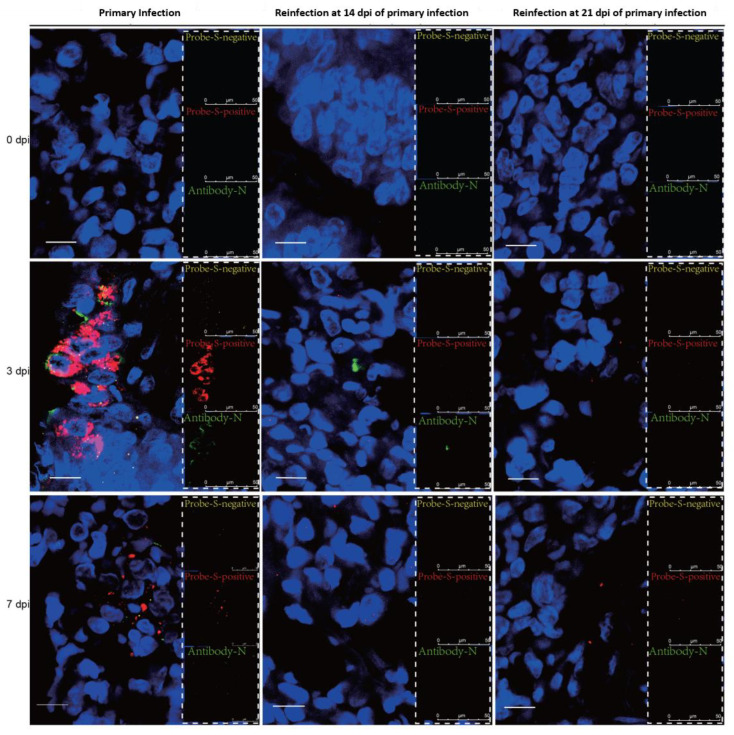
No viral replication was found, and a low viral load was present in the lung during reinfection, while high levels of SARS-CoV-2 viral load and replication were observed during primary infection. The distribution and abundance of SARS-CoV-2 S-mRNA and N-protein in the lung were detected by RNAscope-IF codetection at 0 dpi, 3 dpi, and 7 dpi in the primary and secondary infection groups. Yellow represents the S-mRNA-negative strand, and the signals represent the level of SARS-CoV-2 replication; red represents the S-mRNA-positive strand, and the signals represent the level of SARS-CoV-2 viral load; green represents the N-protein, and the signals represent the level of SARS-CoV-2 viral load. The scale is 10 µm.

**Figure 7 viruses-14-01608-f007:**
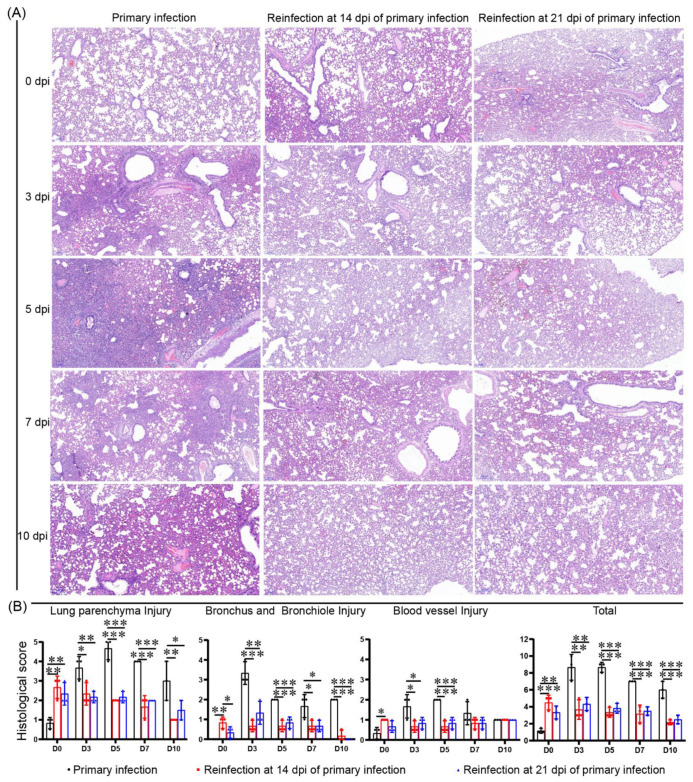
No enhanced pathological changes in the lung were observed during the reinfection of hamsters recovering from severe lung damage from the primary infection. (**A**) Pathological damage to the lung after SARS-CoV-2 primary infection and reinfection was detected by HE staining. (**B**) The scoring of histopathological changes to quantify lung injury, including pulmonary parenchyma, bronchus and bronchiole, and pulmonary vasculature. The histology score standards are shown in Appendix A. *n* = 3 at every point, the data were analyzed using GraphPad Prism 8, and the *p*-values were calculated by one-way ANOVA using SPSS PASW statistical software version 18.0. * 0.01 < *p* ≤ 0.05, ** 0.001 < *p* ≤ 0.01, and *** *p* ≤ 0.001.

**Figure 8 viruses-14-01608-f008:**
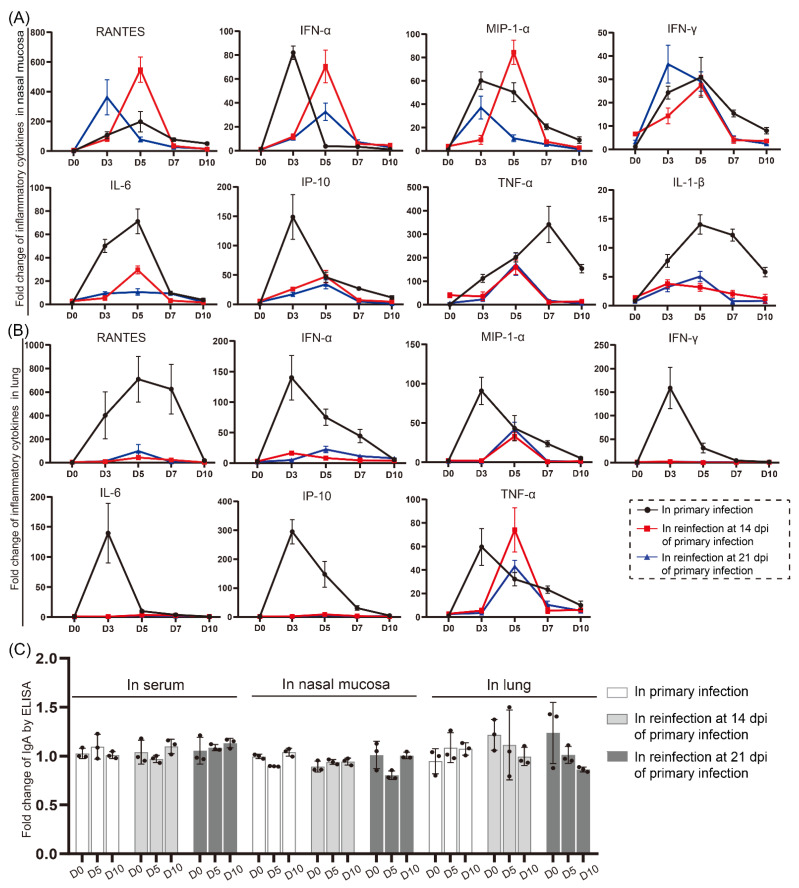
Cytokine/chemokine responses in the nasal mucosa during reinfection were similar to those during primary infection. (**A**,**B**) The mRNA expression levels of cytokines and chemokines, including RANTES, IFN-α, MIP-1-α, IFN-γ, IL-6, IP-10, TNF-α, and IL-1-β, were detected by real-time PCR in the nasal mucosa (**A**) and lung (**B**) during SARS-CoV-2 primary infection and reinfection. (**C**) The levels of IgA in the serum, nasal mucosa, and lung during the infection process of primary and secondary infection were detected by ELISA, there was no significant difference between primary infected group and reinfected group calculated by one-way ANOVA using SPSS PASW statistical software version 18.0. “●” in the image represents the data in every hamster, and the bars were calculated by GraphPad Prism 8 to reflect the data uniformity. *n* = 3 at every point.

## Data Availability

All the data generated or analyzed during this study are included in this published article, and the additional files are available from the corresponding author upon reasonable request.

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
