# Peer review of "Nasal Mucosa Exploited by SARS-CoV-2 for Replicating and Shedding during Reinfection"

_viruses, 2022, doi:10.3390/v14081608_

Round 1

Reviewer 1 Report

It is a very important thesis that in SARS-CoV-2 reinfection, replication is made in the upper respiratory tract of the host, and is expressed only as mild or asymptomatic infection. In addition, it is a very important study to understand reinfection by comparing and analyzing the presence of viruses in upper and lower respiratory tract and differences in cytokines secretion.

in the most important part
1. Is there an infectivity of the virus shedding from the upper respiratory tract?
- Cell culture of upper respiratory tract samples
- Infection of upper airway sample to Syrian hamster
2. Is there any difference between the virus that is released from the first infection, such as the morphological difference of the virus that is discharged from the upper respiratory tract after the reinfection?

If you have an experiment with two questions, add them to the discus if not as an estimate.

Author Response

Point 1. Is there an infectivity of the virus shedding from the upper respiratory tract?
- Cell culture of upper respiratory tract samples
- Infection of upper airway sample to Syrian hamster

Response: Thank you very much for your attention and advice. Basically, we had observed the infectivity of shedding from upper respiratory tract in viral titer testing experiment, in which we found cell cytopathic effects in Vero cells with high titers levels as Figure 1D. In this revision, we supplemented the results of healthy Syrian hamsters infected by nasal washes, and observed increasing viral loads in the nasal mucosa and lung and weight loss, as showed in Figure S2. And the pathological damage in the lung were similar with the primary infection (not shown). We also revised the related description in the Methods, Results and Discussion of this version.

Point 2. Is there any difference between the virus that is released from the first infection, such as the morphological difference of the virus that is discharged from the upper respiratory tract after the reinfection?

Response: Thank you very much for your advice. We didn’t observe morphological changes in viral particle by TEM in the nasal mucosa (as shown in Figure 3B), and in this revision, we observed no morphological difference by TEM in the nasal washes from primary infection and reinfection (as shown in Figure S3), also not in observing of CPE in the plaque experiment (not shown). And we sequenced the SARS-CoV-2 S-RBD gene in the shedding from the upper respiratory tract, and we found no difference in the S-RBD gene sequence from the nasal washes between the primary infection and reinfection (not shown). And we have supplemented them in details in Discussion and Figure S3.

In addition, the Language Editing has been arranged by AJE (www.aje.com).

Reviewer 2 Report

In their manuscript titled “Nasal mucosa exploited by SARS-COV-2 for replicating and 2 shedding during reinfection”, Lia et al. describe experiments in which they repeatedly infected Syrian hamsters with the virus and documented virus replication and pathogenesis.  While the manuscript contains valid data, it is mostly of confirmatory nature; papers by Mohandas et al. (Viruses 2022) and Hansen et al (Cell Rep 2022), cited by the authors, describe the phenomenon in detail.  The present manuscript provides only incremental improvement over those papers. 

Minor points:

Fig. 1c should be converted to bar graph.

The difference between gRNA and sgRNA and their use to indicate virus replication should be explained.

Fig. S2, Table S1, and Fig. S4 are unnecessary.

Fig. 2 should be replaced by Fig. S3.

Author Response

In their manuscript titled “Nasal mucosa exploited by SARS-COV-2 for replicating and 2 shedding during reinfection”, Lia et al. describe experiments in which they repeatedly infected Syrian hamsters with the virus and documented virus replication and pathogenesis.  While the manuscript contains valid data, it is mostly of confirmatory nature; papers by Mohandas et al. (Viruses 2022) and Hansen et al (Cell Rep 2022), cited by the authors, describe the phenomenon in detail.  The present manuscript provides only incremental improvement over those papers. 

Thank you very much for your time and advice. The reinfections in hamsters were indeed described in previous papers, and the shedding in the upper respiratory tract was validated. And comparing with previous papers, in our manuscript, we demonstrated that nasal mucosa exploited by SARS-COV-2 for replicating and shedding during reinfection. In addition, comparing and analyzing the presence of viruses in upper and lower respiratory tract and differences in cytokines secretion were validated in our research. These results would be useful in the immunological and protective evaluation in the infection mechanism research and vaccine development. In addition, the Language Editing has been arranged by AJE (www.aje.com).

Minor points:

Point 1: Fig. 1c should be converted to bar graph.

Response: Thank you very much for your advice, and the results in Fig. 1c have been converted to bar graph.

Point 2: The difference between gRNA and sgRNA and their use to indicate virus replication should be explained.

Response: Thank you very much for your attention and advice. gRNA was positive-sense genomic RNA, sgRNA was subgenomic mRNAs. In the detection, sgRNA and the corresponding gRNA share the same probe and reverse primer (located in a body gene) but differ in the forward primers. The sgRNA-targeting forward primer is located in the 5’ UTR (upstream of the leader-body TRS junction), whereas the cognate gRNA-targeting forward primer is located upstream of the body TRS and coding region (Telwatte S, and et al: Novel RT-ddPCR assays for measuring the levels of subgenomic and genomic SARS-CoV-2 transcripts. Methods 2022). Importantly, the sgRNA provides evidence of replicative inter­mediates of the virus, rather than residual viral RNA (Perera R, and et al: SARS-CoV-2 Virus Culture and Subgenomic RNA for Respiratory Specimens from Patients with Mild Coronavirus Disease. Emerg Infect Dis 2020), and the replication of SARS-CoV-2 depends on transcription of negative-sense RNA intermediates that serve as the templates for the synthesis of gRNA and multiple different sgRNAs (Telwatte S, and et al: Novel RT-ddPCR assays for measuring the levels of subgenomic and genomic SARS-CoV-2 transcripts. Methods 2022). Interestingly, in our manuscript, the sgRNA levels detected by real time PCR were consistent with the SARS-CoV-2 S-negative levels detected by RNAScope in the nasal mucosa and lung in the primary infection and reinfection (Fig 2, 4, 6), and both the sgRNA and the S-negative strands were present in the SARS-CoV-2 replication. The details were described in the Methods and Discussion.

Point 3: Fig. S2, Table S1, and Fig. S4 are unnecessary.

Response: Thank you very much for your advice, and we have deleted the Fig. S2, Table S1, and Fig. S4.

Point 4: Fig. 2 should be replaced by Fig. S3.

Response: Thank you very much for your advice, and we have replaced Fig. 2 by Fig. S3.

Round 2

Reviewer 2 Report

In their revised manuscript titled “Nasal mucosa exploited by SARS-COV-2 for replicating and 2 shedding during reinfection”, Li et al. describe experiments in which they repeatedly infected Syrian hamsters with the virus and documented virus replication and pathogenesis.  The authors did not change the content of the manuscript; thus, it is still of confirmatory nature.  However, in the Discussion, they mention that they did experiments showing that the virus shed during reinfection is not an escape mutant.  These are interesting, novel data and should be included in the Results section. 

The authors significantly improved the language of the manuscript; however, it would still benefit from editing by a colleague with good English and familiarity with the topic. 

Specific points

Line 284: “few or no sgRNAs could be detected in the upper respiratory tracts”  This is in conflict with Fig. 2 showing ~10e7 copies/g in the nasal mucosa.  It needs to be reconciled.

Fig. 2 and the text referring to it mentions the “Nose”.  Define what you mean by it.

Table S1 is jumbled and unreadable.

Line 392 and further: “abundances of cytokines and chemokines”  Make it clear that you are talking about mRNA levels and not actual levels of cytokines and chemokines.

Lines 441-445:  This sentence should be moved to the Results section describing this experiment.

Author Response

In their revised manuscript titled “Nasal mucosa exploited by SARS-COV-2 for replicating and 2 shedding during reinfection”, Li et al. describe experiments in which they repeatedly infected Syrian hamsters with the virus and documented virus replication and pathogenesis.  The authors did not change the content of the manuscript; thus, it is still of confirmatory nature.  However, in the Discussion, they mention that they did experiments showing that the virus shed during reinfection is not an escape mutant.  These are interesting, novel data and should be included in the Results section. 

Thank you very much for your advice. In TEM study, we did not observe obviously morphological changes of viral particles from nasal shedding in reinfection (Figure 1E), and there are no sequence differences in the S1 gene between isolated virus from primary infection and reinfection (Figure S3). We have supplemented these data in the result part of this revision.

The authors significantly improved the language of the manuscript; however, it would still benefit from editing by a colleague with good English and familiarity with the topic. 

Thank you very much for your advice, and in the revision, the language of the manuscript have been checked seriously by 3 of our colleagues.

Specific points

 Line 284: “few or no sgRNAs could be detected in the upper respiratory tracts” This is in conflict with Fig. 2 showing ~10e7 copies/g in the nasal mucosa.  It needs to be reconciled.

Thank you very much for your advice, and in our manuscript, we found that few or no sgRNAs while some gRNAs could be detected in the upper respiratory tracts at 14 and 21 dpi in the primary infection, and the gRNA level was ~10e7 copies in 1 g nasal mucosa at 10 dpi in the primary infection. In details, the viral loads at 14 and 21 dpi in the primary infection were shown as ones at 0 dpi of the two reinfection groups in Figure 2. At 14 dpi of primary infection, the sgRNA level in 1 g nose was 10e1.5 while no sgRNA in the nasal mucosa was detected; at 21 dpi of primary infection, no sgRNA in the nose and nasal mucosa was detected. In the same time, some gRNAs appeared in the upper respiratory tracts at 14 and 21 dpi in the primary infection. In details, at 14 dpi of primary infection, the gRNA level in 1 g nose was 10e4.82 while the gRNA level in 1 g nasal mucosa was 10e5.33; at 21 dpi of primary infection, the gRNA level in 1 g nose was 10e4.6 while the gRNA level in 1 g nasal mucosa was 10e4.52. These were consistent with previous results and no infectious virus was detected at 14 and 21 dpi in the primary infection, and we have described in details in the Introduction (Line 84-86) and Discussion (Line 448-450).

Fig. 2 and the text referring to it mentions the “Nose”.  Define what you mean by it.

Thank you very much for your advice, and Nose means the tissues with nares and no cartilage, Nasal mucosa means the mucous tissue attached to cartilage. We have described in details in the revised manuscript.

Table S1 is jumbled and unreadable.

Thank you very much for your advice, and we have adjusted the word size in the Table S1 in the revised manuscript.

Line 392 and further: “abundances of cytokines and chemokines” Make it clear that you are talking about mRNA levels and not actual levels of cytokines and chemokines.

Thank you very much for your advice, and we have demonstrated the levels of cytokines and chemokines were shown in mRNA levels in the revised manuscript.

Lines 441-445:  This sentence should be moved to the Results section describing this experiment.

Thank you very much for your advice, and we have moved the description into the Results.